# Glycyrrhizin Attenuates Carcinogenesis by Inhibiting the Inflammatory Response in a Murine Model of Colorectal Cancer

**DOI:** 10.3390/ijms22052609

**Published:** 2021-03-05

**Authors:** Guifeng Wang, Keiichi Hiramoto, Ning Ma, Nobuji Yoshikawa, Shiho Ohnishi, Mariko Murata, Shosuke Kawanishi

**Affiliations:** 1Department of Environmental and Molecular Medicine, Mie University Graduate School of Medicine, Tsu, Mie 514-8507, Japan; wangguifeng32948@yahoo.co.jp; 2Sakuranomori Shiroko Home, Social Service Elderly Facilities, Suzuka University of Medical Science, Suzuka, Mie 513-0816, Japan; 3Faculty of Pharmaceutical Sciences, Suzuka University of Medical Science, Suzuka, Mie 513-8670, Japan; hiramoto@suzuka-u.ac.jp (K.H.); shiho-o@suzuka-u.ac.jp (S.O.); 4Graduate School of Health Science, Suzuka University of Medical Science, Suzuka, Mie 513-8670, Japan; maning@suzuka-u.ac.jp; 5Institute of Traditional Chinese Medicine, Suzuka University of Medical Science, Suzuka, Mie 510-0226, Japan; 6Matsusaka R&D Center, Cokey Co., Ltd., Matsusaka, Mie 515-0041, Japan; yoshikawa@cokey.co.jp; 7Graduate School of Pharmaceutical Sciences, Suzuka University of Medical Science, Suzuka, Mie 513-8670, Japan

**Keywords:** colon cancer, Glycyrrhizin, COX-2, HMGB1, 8-NitroG, 8-OxodG, YAP1, SOX9

## Abstract

Glycyrrhizin (GL), an important active ingredient of licorice root, which weakens the proinflammatory effects of high-mobility group box 1 (HMGB1) by blocking HMGB1 signaling. In this study, we investigated whether GL could suppress inflammation and carcinogenesis in an azoxymethane (AOM)/dextran sodium sulfate (DSS)-induced murine model of colorectal cancer. ICR mice were divided into four groups (*n* = 5, each)—control group, GL group, colon cancer (CC) group, and GL-treated CC (CC + GL) group, and sacrificed after 20 weeks. Plasma levels of interleukin (IL)-6 and tumor necrosis factor (TNF)-α were measured using an enzyme-linked immunosorbent assay. The colonic tissue samples were immunohistochemically stained with DNA damage markers (8-nitroguanine and 8-oxo-7,8-dihydro-2′-deoxy-guanosine), inflammatory markers (COX-2 and HMGB1), and stem cell markers (YAP1 and SOX9). The average number of colonic tumors and the levels of IL-6 and TNF-α in the CC + GL group were significantly lower than those in the CC group. The levels of all inflammatory and cancer markers were significantly reduced in the CC + GL group. These results suggest that GL inhibits the inflammatory response by binding HMGB1, thereby inhibiting DNA damage and cancer stem cell proliferation and dedifferentiation. In conclusion, GL significantly attenuates the pathogenesis of AOM/DSS-induced colorectal cancer by inhibiting HMGB1-TLR4-NF-κB signaling.

## 1. Introduction

Inflammatory bowel disease (IBD) is a global healthcare problem, which is experiencing a sustained increase in incidence [1]. It includes two major forms, Crohn’s disease (CD) and ulcerative colitis (UC), which are distinct, chronic, bowel-relapsing inflammatory disorders. CD causes transmural inflammation and can affect any part of the gastrointestinal tract (most commonly, the terminal ileum or the perianal region) in a discontinuous manner. CD is commonly associated with complications such as abscesses, fistulas, and strictures. In contrast, UC is typified by mucosal inflammation and is limited to the colon [2,3]. More recently, it was reported that even without preexisting IBD, inflammation occupies a key position in the development of sporadic colorectal cancer (CRC) [4]. It is well established that inflammatory disorders of the colon are accompanied by an increased risk of developing cancer [5,6].

Licorice (*Glycyrrhiza glabra* and *G. uralensis,* etc.) is an important medicinal plant and its root has been used in traditional medicine for over 2000 years [7,8]. Currently, licorice is found in more than 60% of Japanese traditional medicine (Kampo), and is known to function synergistically with other herbs in the formulas and enhance the efficacy of other ingredients [9]. Pharmacological research has confirmed that licorice has several biologically relevant activities, including anti-oxidative, anti-inflammatory, anti-cancer, anti-viral, immune-regulatory, and hepatoprotective functions [10,11,12,13]. Glycyrrhizin (GL), a triterpene glycoside, is one of the most important active ingredients in licorice, and many biological effects of the plant can be attributed to this compound [7,8,13,14].

High-mobility group box 1 (HMGB1) is a nuclear protein that is released from damaged and necrotic cells [15,16,17]. It plays an important role as a cytokine that triggers inflammation and inflammation-related diseases, including cancer, by upregulating the expression of other inflammatory cytokines [18,19]. GL weakens the proinflammatory effect of HMGB1 by blocking HMGB1 signaling [20,21]. HMGB1 plays important roles in the genesis and promotion of a variety of inflammatory diseases, including different types of cancers [22,23]. There is ample evidence suggesting that GL exhibits its anti-inflammatory effects by inhibiting HMGB1 [24,25,26,27,28,29].

Although some researchers have studied the effect of GL on colitis [30,31,32], the effect of GL on ulcerative colitis colorectal cancer caused by an azoxymethane (AOM)/dextran sodium sulfate (DSS) model is not known. Therefore, we investigated whether GL can suppress inflammation and carcinogenesis, using a carcinogen (AOM)/colorectal inflammatory agent (DSS)-induced murine model of colorectal cancer, and assessed the molecular mechanisms involved.

## 2. Results

### 2.1. Effect of GL on Colon Cancer Induced by AOM/DSS Treatment 

We administered a single intraperitoneal injection of AOM. On the seventh day after the AOM injection, the mice received DSS in drinking water for a week in the CC and CC + GL groups, and then GL (15 mg/kg/day) was administered orally three times a week for 18 weeks in the GL and CC + GL groups. A total of 20 weeks after the experiment started, the presence of colon tumors was scrutinized macroscopically. There were no significant differences in body weight among the four groups (Figure 1a). Length of the colon of the AOM/DSS-induced CC group was significantly shorter than that of the control group (*p* < 0.01, Figure 1b,c). No tumors were observed in the control and GL groups (Figure 1d). The mean number of tumors was 10.0 ± 1.9 in the CC group and 5.8 ± 1.3 in the CC + GL group (Figure 1e). The number of tumors in the AOM/DSS-induced CC group was significantly higher than that in the control group (*p* < 0.01), while GL markedly attenuated tumor formation in the CC + GL group compared with that in the CC group (*p* < 0.05). The mean tumor diameters were 7.4 ± 2.3 and 4.0 ± 1.4 mm in the CC group and CC + GL group, respectively (Figure 1f). GL significantly attenuated the tumor diameter in the CC + GL group compared to that in the CC group (*p* < 0.01).

These observations showed that GL profoundly attenuated tumorigenesis in the murine model of ulcerative colitis-colorectal cancer.

### 2.2. Effects of GL Administration on the Plasma Levels of IL-6 and TNF-α

The levels of inflammatory cytokines, interleukin (IL)-6 and tumor necrosis factor (TNF)-α, in the AOM/DSS-induced CC group were significantly higher than those in the control group (*p* < 0.01), while GL significantly lowered the levels of IL-6 and TNF-α in the CC + GL group compared to those in the CC group (*p* < 0.05; *p* < 0.01, Figure 2a,b).

### 2.3. Histopathological Evaluation of the Effect of GL Administration on the Murine Colonic Epithelia 

Histological examination of hematoxylin and eosin (HE)-stained sections revealed obvious crypt destruction, heterotypic nuclei, and irregular glandular structures in the AOM/DSS-induced CC group (Figure 3c), as compared to the control mice group (Figure 3a) and mice receiving GL group (Figure 3b), which did not exhibit any aberrant features. Most tumors in the CC group exhibited extensive high-grade dysplasia or intra-mucosal carcinoma. In contrast, treatment with GL markedly caused an improvement in crypt structure and reduced tumor formation in AOM/DSS-treated mice (Figure 3d). Most colonic mucosa in the CC + GL group exhibited low-grade dysplasia and much less infiltration of inflammatory cells.

These data suggested that GL could significantly ameliorate colitis-associated colorectal tumorigenesis in mice.

### 2.4. Effects of GL on the Expression of 8-NitroG and 8-OxodG

8-Nitroguanine (8-NitroG) and 8-oxo-7,8-dihydro-2′-deoxy-guanosine (8-OxodG), the well-established DNA damage markers, were observed in the nuclei and cytoplasm of the epithelial cells (brown staining) (Figure 4a,b). Immunohistochemical (IHC) staining of 8-NitroG and 8-OxodG in the control group showed little or no staining, while that in the GL group showed weak staining. The staining in the CC group was higher than that in the control group. On the other hand, staining in the CC + GL group was lower than that observed in the CC group; thus, GL attenuated 8-NitroG and 8-OxodG expression in the CC + GL group.

The IHC score for 8-NitroG in the normal and cancer cells of the colon cancer tissue in the CC group was higher than that in the normal cells of the control group. The IHC score for 8-NitroG in the normal and cancer cells of the cancer tissue in the CC + GL group was lower than that in the CC group (Figure 4c). These results indicate that the AOM/DSS treatment induces the expression of 8-NitroG in colonic cells during carcinogenesis, and GL lowers the expression of 8-NitroG in the normal cells surrounding the cancer tissue.

The IHC score for 8-OxodG in the cancer cells of the cancer tissue in the CC group was higher than that in the control group; however, the score in the CC + GL group was lower than that in the CC group (Figure 4d). These results indicate that AOM/DSS treatment induces the expression of 8-OxodG in colonic cells during carcinogenesis, and GL prevents the expression of 8-OxodG in cancer cells.

### 2.5. Effects of GL on the Expression of COX-2 and HMGB1

Cyclooxygenase (COX)-2, an inflammatory marker, was observed in the cytoplasm of epithelial cells (brown staining) (Figure 5a). The IHC staining of COX-2 in the control and GL groups showed little or no staining. COX-2 expression in the CC group was higher than that in the control group; however, the expression was lower in the CC + GL group than in the CC group. Thus, GL treatment resulted in low COX-2 staining in the CC + GL group compared to that in the CC group.

HMGB1, an inflammatory cytokine, was observed in the nuclei and cytoplasm of epithelial cells (brown staining) (Figure 5b). IHC staining of HMGB1 in the control and GL groups showed a weak staining pattern. The IHC staining of HMGB1 in the CC group was higher than that in the control group, and this increase was prevented in the CC + GL group. GL lowered the IHC staining of HMGB1 in the CC + GL group compared with that in the CC group.

The IHC score for COX-2 in the cancer cells in the CC group was higher than that in the control group. The score in the cancer cells in the CC + GL group was lower than that in the CC group (Figure 5c). These results indicate that AOM/DSS treatment induces the expression of COX-2 in cancer cells during carcinogenesis, and GL prevents the expression of COX-2 in cancer cells.

The IHC score for HMGB1 in the normal and cancer cells of the colon cancer tissue in the CC group was higher than that in the control group. The IHC score for HMGB1 in the normal and cancer cells of the cancer tissue in the CC + GL group was lower than that in the CC group (Figure 5d). These results indicate that the AOM/DSS treatment induces the expression of HMGB1 in colonic cells during carcinogenesis, and GL attenuates the expression of HMGB1 in the normal surrounding the cancer tissue.

### 2.6. Effects of GL on the Expression of YAP1 and SOX9

Yes-associated protein (YAP)1 and sex-determining region Y (SRY)-box (SOX) 9 protein (SOX9) are cancer stem cell markers and were observed in the nuclei of the colonic epithelial cells (brown staining) (Figure 6a,b). IHC staining of YAP1 was showed little staining and SOX9 showed little staining in the control and GL groups. The IHC staining of YAP1 and SOX9 in the CC group was higher than in the control groups; however, staining in the CC + GL group was lower than that observed in the CC group. Thus, GL reduced the IHC staining of YAP1 and SOX9 in the CC + GL group compared to that in the CC group.

The IHC score for YAP1 in the cancer cells of the cancer tissue in the CC group was higher than that in the control group; however, the score in the CC + GL group was lower than that in the CC group (Figure 6c).

The IHC score for SOX9 in the normal and cancer cells of the colon cancer tissue in the CC group was higher than that in the normal cells of the control group. The IHC score for SOX9 in the normal and cancer cells of the cancer tissue in the CC + GL group was lower than that in the CC group (Figure 6d). These results indicate that the AOM/DSS treatment induces the expression of SOX9 in colonic cells during carcinogenesis, and GL attenuates the expression of SOX9 in the normal surrounding the cancer tissue.

## 3. Discussion

GL is main component of the Chinese herbal medicine licorice, and we demonstrated that it attenuates carcinogenesis in an AOM/DSS mouse model. Colon tumors observed after AOM/DSS treatment were in accordance with procedures from previous studies [33,34]. Compared to the AOM/DSS group, the GL-treated group showed an improvement in length of colon and number of tumors and infiltration of inflammatory cytokines in the colon. Histological observation revealed that GL has potent anti-inflammatory properties. These findings together with the histological data highlight the protective effects of GL against AOM/DSS-induced colonic damage. In addition, we found that after GL administration, the plasma levels of IL-6 and TNF-α in the CC + GL group were lower than those in the CC group, suggesting that GL attenuates inflammation in AOM/DSS-induced colitis. This is consistent with the observation that GL can suppress the development of precancerous lesions by regulating hyperproliferation and inflammation in the colon of Wistar rats [32]. This anti-inflammatory activity of GL may be explained by the fact that GL can specifically bind HMGB1 and inhibit its cytokine activity [35,36]. HMGB1 is a recently identified protein associated with cancer growth and metastasis, and represents a new therapeutic target for the treatment of cancer [37]. Moreover, Tripathi et al. have shown that HMGB1 could be used as a marker for the prognosis of tumor stages and can be targeted for cancer therapy. Overexpression of HMGB1 plays an important role in the migration of cells, tumor progression, and metastasis in colorectal cancer; thus, it could be used as a predictor of disease outcome [23]. Our results show that GL inhibits the inflammation-induced carcinogenesis by regulating HMGB1, which is consistent with the carcinogenic theory of inflammation that we have previously advocated [38].

With respect to inflammation-induced DNA damage, we found that expression of 8-NitroG, 8-OxodG, COX-2, and HMGB1 in the cancerous cells of the CC group significantly increased compared to that in the normal cells of the control group, which suggests that DNA damage and inflammatory markers are involved in the induction of cancer. However, GL significantly decreased the expression of all these markers in the cancerous cells of the CC + GL group. On the basis of these results and previous literature, we propose a possible mechanism by which GL attenuates carcinogenesis by inhibiting inflammation in an ulcerative colitis-colorectal cancer mouse model (Figure 7).

First, AOM induces DNA damage, which leads to cell death in addition to accumulation of mutations as a result of the DNA damage response (DDR). HMGB1 is a nuclear protein that is released from dead cells [39]. We have previously shown that indium compounds induced inflammation-mediated DNA damage in lung epithelial cells via the HMGB1 pathway [40]. In addition, by binding to toll-like receptor (TLR) 4, HMGB1 activates myeloid differentiation (MyD) 88 and NF-κB essential modulator (NEMO), and activates inflammatory cytokines such as IL-6 and TNF-α via nuclear factor-kappa B (NF-κB) [16,17,18,41,42,43,44]. HMGB1 induces cytokine release from both recruited leukocytes and resident immune cells, including TNF-α and IL-6, which amplify and extend the inflammatory response [42]. TNF-α and IL-6 are proinflammatory cytokines that play important roles in the control of inducible nitric oxide synthase (iNOS) expression via regulation of the NF-κB and signal transducer and activator of transcription (STAT) 3 signaling pathways. Our previous studies have demonstrated that Epstein–Barr virus infection may induce nuclear accumulation of EGFR and IL-6-induced STAT3, leading to iNOS and NADPH oxidase (Nox) expression. Reactive oxygen species (ROS) and reactive nitrogen species (RNS), generated via these enzymes, can induce the formation of mutagenic DNA lesions, including 8-NitroG and 8-OxodG, respectively [45,46,47,48]. In addition, during inflammation, the signaling cascade stimulates the activation of NF-κB, which induces pro-inflammatory genes, including iNOS and COX-2. COX-2 catalyzes the conversion of arachidonic acid (AA) to prostaglandin (PG) H2, which is converted into PGE2 by the terminal PGE synthase. AA represents the main substrate of COX-2 for PGE2 production [49]. PGE2 transduces signals via four G-protein coupled receptors (EP2) to activate NF-κB. Overexpression of NF-κB promotes the expression of COX-2, leading to DNA damage, which is responsible for the repeated release of HMGB1 [38].

We have previously reported that DNA damage, including the formation of 8-NitroG and 8-OxodG, increases mutagenesis and genomic instability, finally leading to carcinogenesis [38,50]. In addition, in a mouse model of IBD, we demonstrated the accumulation of 8-NitroG and 8-OxodG in colonic epithelial cells [51]. The accumulation of DNA lesions was related to the expression of iNOS and proliferating cell nuclear antigen (PCNA). These results suggest that iNOS-dependent DNA damage is induced in the colonic epithelial cells of the AOM/DSS mouse model, which may lead to cell proliferation and carcinogenesis.

Knowledge of the expression pattern of cancer stem cells (CSCs) in CRC has been increasing in recent years, revealing a heterogeneous population of cells within CRC, ranging from pluripotent to differentiated cells, with overlapping and sometimes unique combinations of markers [52]. In addition to stem and progenitor cells, CSCs have been shown to arise from more differentiated cells as a consequence of constitutive NF-κB activation and chemically-induced inflammation in CRC [53]. We have previously reported that increased DNA damage due to inflammation may result in the mutation of stem cells, leading to tumor development [54]. It has been reported that NF-κB is activated by YAP1 [55,56]. To determine whether CSCs involved in inflammation can also cause tumor development, we examined the expression of stem cell markers YAP1 and SOX9, and found that their expression in the cancer cells in the CC group was significantly higher than that in the normal cells of the control group. Interestingly, GL significantly decreased the levels of YAP1 and SOX9. In addition, SOX9 levels were significantly lower in the normal cells surrounding the cancer tissues in the CC + GL group, which suggests that GL may affect stemness by attenuating inflammation.

Qian et al. showed that hyperactivation of HMGB1-RAGE signaling contributes to CSCs in CRC development [57]. HMGB1 and DSS promote the release of inflammatory cytokines such as IL-6 and TNF-α [58], as shown in Figure 7. IL-6 promotes the survival of intestinal epithelial cells [59]. Mucosal regeneration after a DSS challenge requires concomitant activation of YAP1 [60]. IL-6 is released from macrophages after mucosal injury in the intestine, and its direct effectors are Janus kinase (JAK)-STAT3 and YAP [61]. Recently, a new network was discovered between IL-6 and YAP1, which led to an increase in colonic tumor formation [61,62,63,64]. IL-6–gp130 signaling has also been shown to activate YAP [65], which in turn, promotes IL-6-induced STAT3 phosphorylation and NF-κB activation [66,67,68,69,70].

Consequently, YAP1 has gained considerable attention as a critical mediator involved in the expansion of CSCs and inhibition of their differentiation [71,72]. YAP mediates its function by binding TAZ (transcriptional co-activator with PDZ-binding motif). In tumors, YAP/TAZ can reprogram cancer cells into CSCs and induce tumor initiation, progression, and metastasis [73,74]. In the intestine, expression of endogenous YAP1 is restricted to the progenitor/stem cell compartment, and activation of YAP1 expands multipotent undifferentiated progenitor cells, which express specific transcription factors such as TEA-domain (TEAD) factors [75]. YAP controls genes that stimulate cell proliferation and tissue growth and inhibit terminal differentiation [76]. It also activates its downstream target SOX9 via TEAD1. Wang et al. observed a positive correlation between YAP signaling and SOX9 in esophageal squamous cell carcinoma [77]. SOX9 is a marker of stem cells and is a regulator of YAP1 signaling [77,78]. As SOX9 is an oncogene, its upregulation is common in colorectal adenomas and cancer and is an independent indicator of the poor prognosis of CRC [79]. Thus, CSCs are formed not only by NF-κB and DNA damage but also via YAP1.

Thus, we conclude that GL attenuated the carcinogenesis in an AOM/DSS-induced colorectal cancer model. The high levels of COX-2 and HMGB1 promoted inflammation and DNA damage, marked by 8-NitroG and 8-OxodG, and the dedifferentiation of cancer cells into YAP1- and SOX9-positive CSCs in the colonic tissue. All these markers were significantly suppressed by GL. Based on our results, we propose a new GL-based mechanism for the prevention of carcinogenesis in the colonic cells of mice treated with AOM/DSS.

## 4. Materials and Methods

### 4.1. Animals and Chemicals

For this study, 8-week-old female ICR mice were purchased from Japan SLC Inc. (Hamamatsu, Japan). This study was conducted in accordance with the recommendations of the Guide for the Care and Use of Laboratory Animals of Suzuka University (approval number—34). All surgeries were performed under pentobarbital anesthesia and efforts were made to minimize animal suffering. Mice were acclimated for 1 week with tap water and a pelleted diet, ad libitum, before the start of the experiment. They were housed under controlled conditions of humidity (50 ± 10%), light (12/12 h light/dark cycle), and temperature (22 ± 2 °C).

The colonic carcinogen AOM was purchased from Sigma Chemical Co. (St. Louis, MO, USA). DSS with a molecular weight of 36,000–50,000 was purchased from MP Biomedicals, Inc. (Solon, OH, USA), and GL (>98%) was purchased from Nagara Science Co., Ltd. (Gifu, Japan).

### 4.2. Experimental Procedure

The experimental protocol of this study is outlined in Figure 8. The mice were quarantined for the first 7 days and then randomized according to bodyweight into four groups (*n* = 5, each). Group 1 (control)—the mice were intraperitoneally injected with saline and given distilled water (DW) for 20 weeks. Group 2 (GL)—the mice were administered DW for 2 weeks after the initial intraperitoneal saline injection, and then approximately 15 mg/kg/day of GL dissolved in phosphate-buffered saline (PBS) (pH 7.4) was administered orally three times a week for 18 weeks. Group 3 (CC)—mice were given a single intraperitoneal injection of AOM (10 mg/kg body weight). Starting 1 week after the injection, the animals received 2% DSS in their drinking water for 7 days and no further treatment for 18 weeks in accordance with previously described procedures [33,80,81]. Group 4 (CC + GL)—the mice were administered AOM/DSS as in the CC group, and GL (15 mg/kg/day, orally) three times per week for 18 weeks. Bodyweight was checked twice a week after DSS treatment. All animals were sacrificed using pentobarbital at the end of the study (week 20). For plasma preparation, blood was collected from the heart into heparinized tubes before the autopsy. During the autopsy, the large bowel was flushed with saline and excised. The large bowel (from the ileocecal junction to the anal verge) was measured, dissected longitudinally along the main axis, and then washed with saline. The tumor lesions were counted by two investigators.

### 4.3. Quantification of IL-6 and TNF-α Levels

Plasma was obtained from the blood samples by centrifugation at 3000× *g* for 10 min at 4 °C and used for analysis. Plasma levels of IL-6 and TNF-α were measured using commercial enzyme-linked immunosorbent assay kits (BioLegend, San Diego, CA, USA), according to the manufacturer’s instructions.

### 4.4. Histopathological and Immunohistochemical Studies

Colonic tissue samples were fixed in 4% formaldehyde in PBS for one day. Following dehydration and paraffin infiltration, the tumors were embedded in paraffin blocks and sectioned to 6 µm thickness using a Leica RM2265 Microsystems (Wetzlar, Germany) by routine procedures. The histopathological appearance of the mouse tumors was evaluated by staining with HE staining. Benign and malignant lesions were histopathologically distinguished using HE-stained samples by two investigators.

For IHC analysis, paraffin-embedded mouse colon sections were deparaffinized in xylene and hydrated in a series of alcohols. After heat-induced antigen retrieval and blocking with 1% skim milk, the sections were incubated overnight with primary antibodies (8-NitroG (Pinlaor et al., 2004 [31], 1:400)); 8-OxodG (JaICA, MOG-100p, 1:400); HMGB1 (Abcam, 18256, 1:400); COX2 (Santa Cruz Biotechnology, Inc., SC-1745, 1:400); YAP1 (Abcam, ab 39361, 1:400); SOX9 (Abcam, ab 185230, 1:400)), and then incubated with an avidin–biotin complex (Vectastain ABC kit, Vector Laboratories Burlingame, CA, USA). The immunoreaction was visualized using a peroxidase DAB kit (Nacalai Tesque Inc., Kyoto, Japan). The tissues were observed and imaged under a microscope (BX51, Olympus, Tokyo, Japan). The semiquantitative analysis of staining intensity was graded by an IHC score between 0 and 4 by two investigators as follows—no staining (0), weak staining (1+), moderate staining (2+), strong staining (3+), and very strong staining (4+).

### 4.5. Statistical Analysis

Comparison of data between groups was analyzed using the Mann–Whitney U test using SPSS. A *p*-value of less than 0.05 was considered statistically significant. SPSS results after statistical analysis were plotted using Graphpad Prism8.

## Figures and Tables

**Figure 1 ijms-22-02609-f001:**
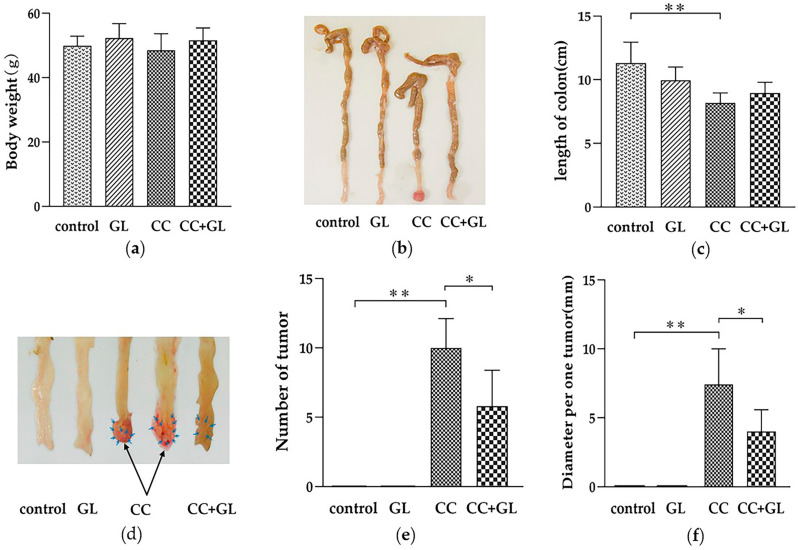
Effect of glycyrrhizin (GL) administration on colon cancer induced by azoxymethane (AOM) and dextran sodium sulfate (DSS)—(**a**) body weight; (**b**) typical colon samples from each group after dissection (from the ileocecal junction to the anal verge); (**c**) length of colon and comparison between the four groups; (**d**) tumors (arrows) formed in the colon; (**e**) number of tumors in the colon and comparison between the four groups; and (**f**) tumor diameter and comparison between the four groups. * *p* < 0.05; ** *p* < 0.01.

**Figure 2 ijms-22-02609-f002:**
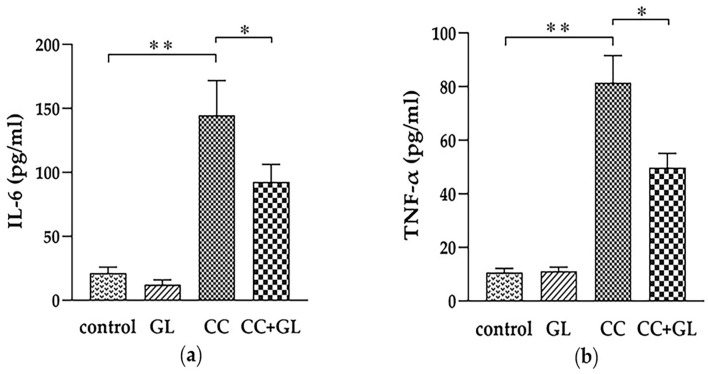
Effects of GL on the plasma levels of interleukin 6 (IL-6) (**a**) and tumour necrosis factor α (TNF-α) (**b**). * *p* < 0.05; ** *p* < 0.01.

**Figure 3 ijms-22-02609-f003:**
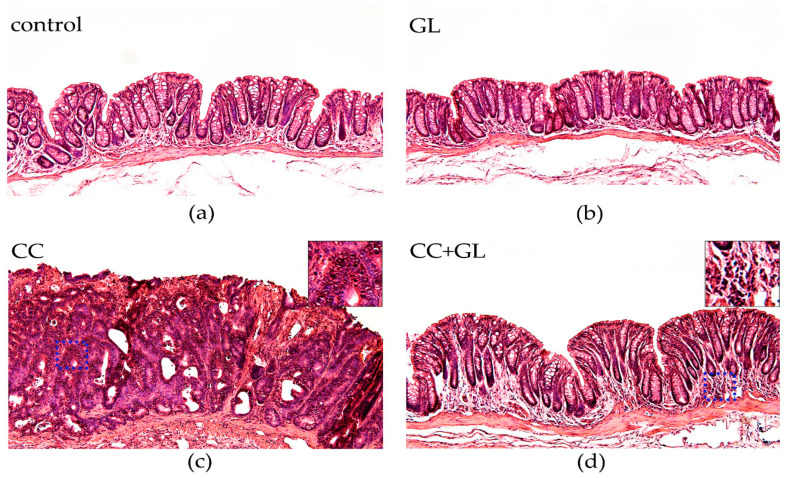
Microscopic examination of murine colonic tissues with hematoxylin and eosin (HE) staining. Representative histological sections of (**a**) control group; (**b**) GL group; (**c**) CC group; and (**d**) CC + GL group. Original magnification—100×.

**Figure 4 ijms-22-02609-f004:**
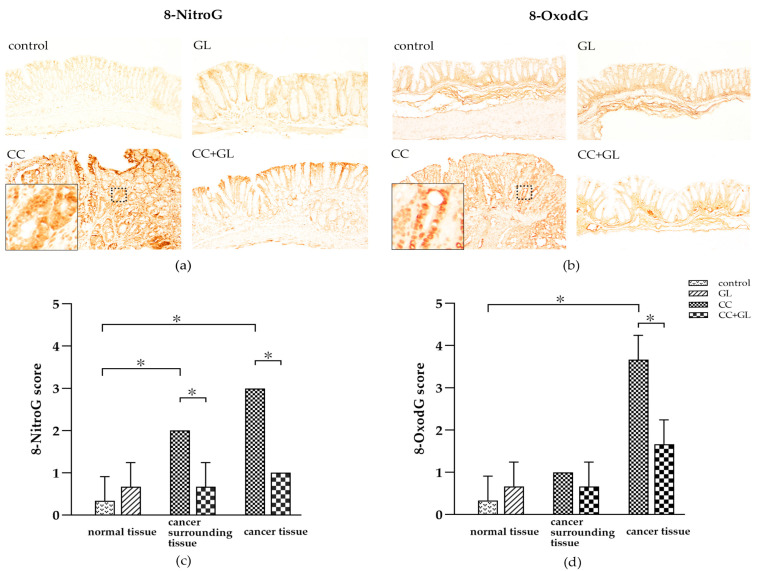
Immunohistochemical (IHC) staining for (**a**) 8-NitroG and (**b**) 8-OxodG in the colonic tissues of the four groups of mice. Brown color indicates specific immunostaining. Cancer surrounding tissue represents the normal cells adjacent to the colon cancer tissue. Original magnification—100×. IHC score for (**c**) 8-NitroG and (**d**) 8-OxodG in the colonic tissues of the four groups of mice. Graphs represent the average score (bar: SD; * *p* < 0.05).

**Figure 5 ijms-22-02609-f005:**
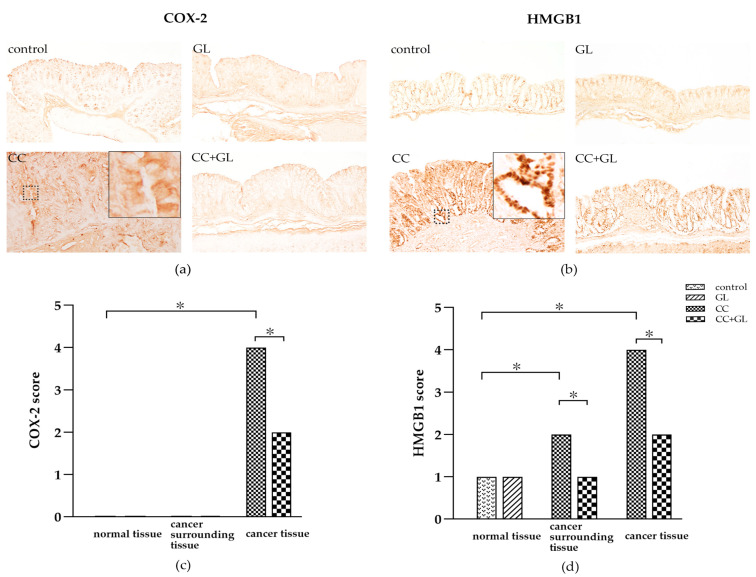
IHC staining of (**a**) cyclooxygenase (COX)-2 and (**b**) high-mobility group box 1 (HMGB1) in the colonic tissues of the four groups of mice. Brown color indicates specific immunostaining. Original magnification—100×. IHC score for (**c**) COX-2 and (**d**) HMGB1 in the colonic tissues of the four groups of mice. Graphs represent the average score (bar: SD; * *p* < 0.05).

**Figure 6 ijms-22-02609-f006:**
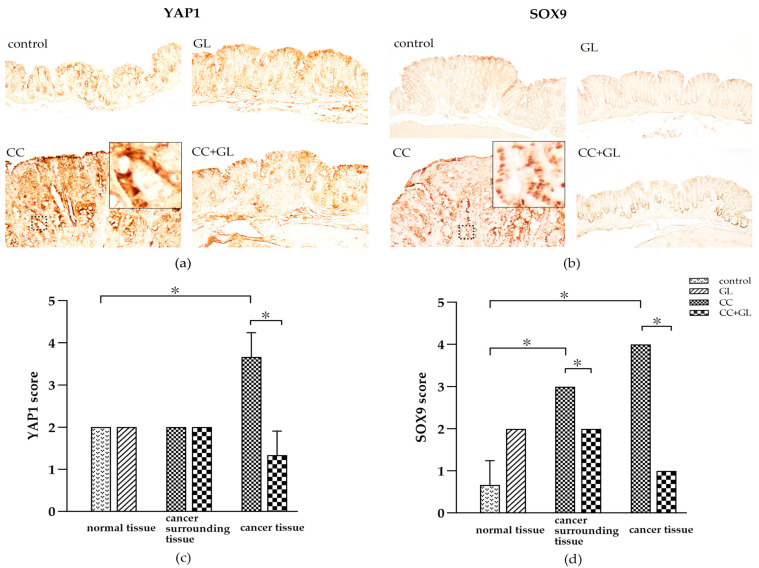
IHC staining of (**a**) yes-associated protein (YAP) 1 (YAP1) and (**b**) sex-determining region Y (SRY)-box (SOX) 9 (SOX9) in the colonic tissues of the four groups of mice. Brown color indicates specific immunostaining. Original magnification—100×. IHC score for (**c**) YAP1 and (**d**) SOX9 in the colonic tissues of the four groups of mice. Graphs represent the average score (bar: SD; * *p* < 0.05).

**Figure 7 ijms-22-02609-f007:**
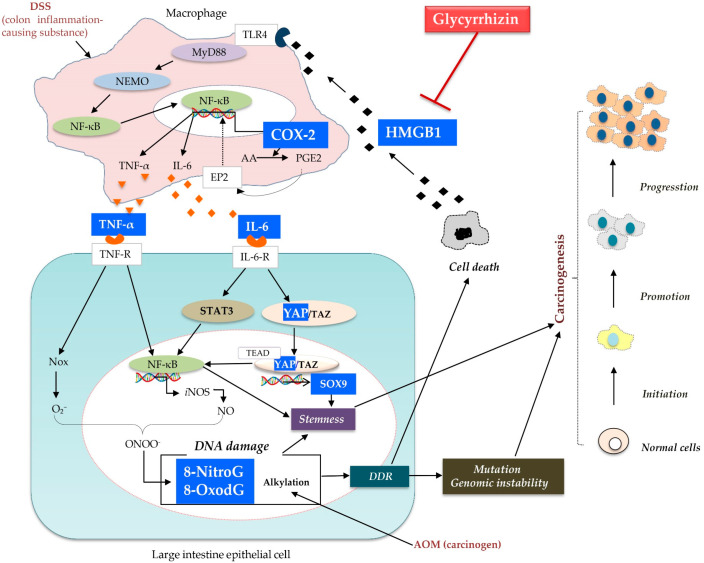
A possible mechanism of action of glycyrrhizin (GL) in colon cancer. GL inhibits the binding of its target protein HMGB1 to the toll-like receptor 4 (TLR4) receptor and blocks the downstream MyD88-NEMO pathway. Subsequently, NF-κB nuclear translocation induces the pro-inflammatory factors IL-6 and TNF-α. GL also suppresses COX-2 expression. In addition, the inhibition of inflammatory cytokines IL-6 and TNF-α lowers the expression of downstream DNA damage markers including 8-NitroG and 8-OxodG. IL-6-induced cancer stem cell markers, YAP1 and SOX9, are also inhibited by GL. Therefore, GL attenuates carcinogenesis by inhibiting inflammation in ulcerative colitis-colorectal cancer. Note—pathways are simplified and only key elements are shown.

**Figure 8 ijms-22-02609-f008:**
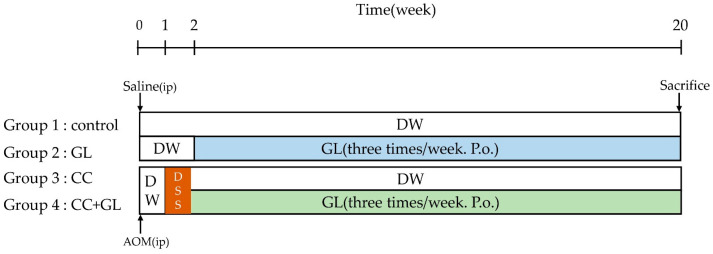
Experimental protocol.

## Data Availability

The data presented in this study are available in the article.

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
