# Peer review of "Glycyrrhizin Attenuates Carcinogenesis by Inhibiting the Inflammatory Response in a Murine Model of Colorectal Cancer"

_ijms, 2021, doi:10.3390/ijms22052609_

Round 1

Reviewer 1 Report

Comments to the Authors

The manuscript from Guifeng Wang et al. entitled “Glycyrrhizin Attenuates Carcinogenesis by Inhibiting the Inflammatory Response in a Murine Model of Colorectal Cancer” seems to be an interesting manuscript. There are points which need to be addressed.

  1. Lines 77-, please check if a plus sign, such as “CC+GL” and “AOM+DSS,” can be used in English paper.
  2. Line 88, a unit for diameter should be described.
  3. Figure 1, reviewers do not see there are 10 tumors with an average diameter of 7-8 millimeters in the CC group from the picture of Figure 1d. Authors can display the pictures of multiple large intestines per group.
  4. Line 198, what are colon cancer symptoms?
  5. Lines 221-, this does not seem an explanation of the mechanism. Markers are only indicators, and decreasing them cannot be a mechanism.
  6. Figure 7, no mention of hypoxia in the manuscript and legend. Is that important and necessary for this study?
  7. Line 321, the age and gender of mice are not same as the previous study cited by authors (#33). Any reason?
  8. Lines 336-, please check if the mice in Group 1 were injected with DW as well as saline.
  9. Line 338, “distilled water” can be “DW” as shown previously.
  10. Line 343, authors can show and cite previous paper(s) describing the procedures.
  11. Line 336, please check if authors used only “tumor sections” for IHC analysis.
  12. Figure 3c may include tumor cells. How was this judged as colitis? And who did? Also, in Figures 4 and 5, which cells are stained by antibodies, especially COX-2? Authors should add enlarged pictures. Overall, this manuscript shows many HE and IHC pictures and they are important for suggested mechanisms; therefore, if not, specialist(s) for pathology should be involved as co-author and evaluate the tissue and tumors of mice colorectum.

Author Response

Reviewer #1

The manuscript from Guifeng Wang et al. entitled “Glycyrrhizin Attenuates Carcinogenesis by Inhibiting the Inflammatory Response in a Murine Model of Colorectal Cancer” seems to be an interesting manuscript. There are points which need to be addressed.

Answer: We thank you for the critical reading. We would like to answer, point by point, as below.

Lines 77-, please check if a plus sign, such as “CC+GL” and “AOM+DSS,” can be used in English paper.

Answer: English correction has done by an English-proofreading company, Editage, as the attached certification. Some papers use like “CC+GL” for the treated group names, so, we would like to use “CC+GL”. However, many papers use AOM/DSS, but not AOM+DSS, as your comment. So, we changed “AOM+DSS” to “AOM/DSS” throughout the manuscript.

Line 88, a unit for diameter should be described.

Answer: We added the unit “mm”.

Figure 1, reviewers do not see there are 10 tumors with an average diameter of 7-8 millimeters in the CC group from the picture of Figure 1d. Authors can display the pictures of multiple large intestines per group.

Answer: According to your suggestion, we added an image for CC group with multiple tumors of colon. In addition, we pointed tumors by arrows.

Line 198, what are colon cancer symptoms?

Answer: Thank you for your insightful comment. What we wanted to say was “length of colon, number of tumors, and infiltration of inflammatory cytokines in colon”. Therefore, we changed.

Lines 221-, this does not seem an explanation of the mechanism. Markers are only indicators, and decreasing them cannot be a mechanism.

Answer: As you suggested, our explanation of the mechanism may be too speculative because of the results from IHC. Therefore, we deleted several sentences (lines 211 – 218 in the original manuscript). Also, we changed the title of Fig. 7 to “A possible mechanism …”.

Figure 7, no mention of hypoxia in the manuscript and legend. Is that important and necessary for this study?

Answer: We deleted the part of hypoxia in Fig. 7.

Line 321, the age and gender of mice are not same as the previous study cited by authors (#33). Any reason?

Answer: We performed a preliminary study using 8-week-old female mice for AOM/DSS protocol as same as reference 33, and got a good result, as colorectal cancer model. Furthermore, female mice are easier to handle than male mice.

Lines 336-, please check if the mice in Group 1 were injected with DW as well as saline.

Answer: Thank you for the careful reading. The mice in the control group were intraperitoneally injected with saline, and given drinking DW for 20 weeks.

Line 338, “distilled water” can be “DW” as shown previously.

Answer: We corrected it.

Line 343, authors can show and cite previous paper(s) describing the procedures.

Answer: We cited three papers.

Line 336, please check if authors used only “tumor sections” for IHC analysis.

Answer: We corrected it as “colon sections”.

Figure 3c may include tumor cells. How was this judged as colitis? And who did? Also, in Figures 4 and 5, which cells are stained by antibodies, especially COX-2? Authors should add enlarged pictures. Overall, this manuscript shows many HE and IHC pictures and they are important for suggested mechanisms; therefore, if not, specialist(s) for pathology should be involved as co-author and evaluate the tissue and tumors of mice colorectum.

Answer: Thank you for your valuable comment. The author, Prof. Ning Ma (MD, PhD) is the specialist for anatomy and histopathology. We modified the sentences in the section 2.3 (H&E) to be more clearly. In addition, we replaced the pictures of Fig. 3 – 6 with clearer images with enlarged insets.

Reviewer 2 Report

It is well-known that chronic inflammation induces colon tumor growth. In this study Wang et al. show that an antiinflammatory compound, glycyyrhizin, attenuates carcinogenesis in a mouse colorectal cancer model. Glycyrrhizin inhibits HMGB1 and lower the levels of inflammtory markers in both plasma and colon tissue. Further, glycyrrhizin down-regulates the expression of DNA damage markers and stem cell markers. In glycyrrhizin trated mice both number and size of tumors was reduced by GL. 
In conclusion, Wang et al. propose a new orally adminstred GL mediated mechanism for the prevention of colon cacer. 

I have only some minor corrections:
- line 174: (SOX) 9 are
- Abbreviations: STAT?

Author Response

Reviewer #2

It is well-known that chronic inflammation induces colon tumor growth. In this study Wang et al. show that an antiinflammatory compound, glycyyrhizin, attenuates carcinogenesis in a mouse colorectal cancer model. Glycyrrhizin inhibits HMGB1 and lower the levels of inflammtory markers in both plasma and colon tissue. Further, glycyrrhizin down-regulates the expression of DNA damage markers and stem cell markers. In glycyrrhizin trated mice both number and size of tumors was reduced by GL.

In conclusion, Wang et al. propose a new orally adminstred GL mediated mechanism for the prevention of colon cacer.

I have only some minor corrections:

- line 174: (SOX) 9 are

- Abbreviations: STAT?

Answer: Thank you for your kind suggestions.

We corrected SOX9 and STAT abbreviation.